# Climate-driven shifts in sediment chemistry enhance methane production in northern lakes

E.J.S. Emilson [1,4], M.A. Carson [2], K.M. Yakimovich [2], H. Osterholz[3], T. Dittmar[3], J.M. Gunn[2], N.C.S. Mykytczuk[2], N. Basiliko[2] & A.J. Tanentzap[1]

Freshwater ecosystems are a major source of methane ($CH_4$), contributing 0.65 Pg (in $CO_2$ equivalents) $yr^{-1}$ towards global carbon emissions and offsetting ~25% of the terrestrial carbon sink. Most freshwater $CH_4$ emissions come from littoral sediments, where large quantities of plant material are decomposed. Climate change is predicted to shift plant community composition, and thus change the quality of inputs into detrital food webs, with the potential to affect $CH_4$ production. Here we find that variation in phenol availability from decomposing organic matter underlies large differences in $CH_4$ production in lake sediments. Production is at least 400-times higher from sediments composed of macrophyte litter compared to terrestrial sources because of inhibition of methanogenesis by phenol leachates. Our results now suggest that earth system models and carbon budgets should consider the effects of plant communities on sediment chemistry and ultimately $CH_4$ emissions at a global scale.

[1] Ecosystems and Global Change Group, Department of Plant Sciences, University of Cambridge, Downing St., Cambridge CB2 3EA, United Kingdom. [2] Vale Living with Lakes Centre, Laurentian University, 935 Ramsey Lake Rd., Sudbury, ON P3E 2C6, Canada. [3] ICBM-MPI Bridging Group for Marine Geochemistry, Institute for Chemistry and Biology of the Marine Environment, Carl von Ossietzky University Oldenburg, Carl-von-Ossietzky-Straße 9-11, 26129 Oldenburg, Germany. [4] Present address: Natural Resources Canada, Great Lakes Forestry Centre, 1219 Queen St. E., Sault Ste. Marie, ON P6A 2E3, Canada. Correspondence and requests for materials should be addressed to E.J.S.E. (email: erik.emilson@canada.ca)

Lentic freshwater ecosystems are a major source of methane ($CH_4$), contributing 0.65 Pg (in $CO_2$ equivalents) $yr^{-1}$ towards global carbon (C) emissions and accounting for an estimated 6–16% of natural $CH_4$ emissions as compared to 1% from the oceans[1]. Freshwater $CH_4$ emissions are enough to offset an estimated ~25% of the terrestrial carbon sink in $CO_2$ equivalents[2]. Within individual lakes, up to 77% of $CH_4$ emissions can come from production in littoral sediments, where warm temperatures and accumulated organic matter (OM) promote methanogen activity and ebullition[3–5], and shallow waters and wave action facilitate rapid diffusion[6,7].

In northern (temperate and boreal) lakes, which account for most of the planet's ice-free freshwater[8,9], rates of $CH_4$ emission from littoral sediments are known to vary by at least three orders of magnitude[3], leaving considerable uncertainty to be explained in regional and global C budgets. In general, emissions are highest where littoral zones are covered with macrophytes[3], and plant-related $CH_4$ fluxes remain one of the least-understood components of the global methane budget[10]. Emergent aquatic plants can directly transport $CH_4$ to the atmosphere through aerenchyma cells, but this cannot explain all of the variability observed within vegetated littoral zones[7,11,12], nor can differences in sediment temperature and OM content[13]. Another explanation is that the activity of sediment microbial communities is inhibited, to varying degrees, by the breakdown of different OM sources[14], resulting in variation in the production of $CH_4$ in littoral sediments. Therefore, regional estimates of $CH_4$ emissions may need to consider the aerial coverage of different plant species and functional types if they contribute OM that differentially influences rates of sediment $CH_4$ production.

Water-soluble phenolic compounds from plant litter have specifically been shown to bind to and inactivate extracellular enzymes and exert toxicity in methanogens[15,16]. These compounds build-up in anaerobic soils and sediments because oxygen limitation restricts phenol oxidase activity and dark conditions prevent photodegradation[15,17]. In this way, the buildup of phenolic compounds may act similar to a 'latch', suppressing $CH_4$ production and holding in place large quantities of C in lake sediments that would otherwise be released as $CH_4$. Oxygen limitation plays a similar role in sequestering $CO_2$ in peatlands by restraining phenol oxidase activity[17], and rates of $CH_4$ production have been related to peat chemical composition[18–20].

Here we show that the production of $CH_4$ in northern lakes can vary by at least 400-times because of differences in sediment chemistry related to sources of plant litterfall. We predicted that sediments would differ in concentration of methanogenesis-inhibiting phenols according to incoming sources of OM. To test the effects of these differences in sediment chemistry on $CH_4$ production in lakes, we compared natural sediments amended with OM from three widespread sources in north-temperate watersheds that vary in phenol content (Supplementary Table 1): mixed coniferous forest litter (CON), mixed deciduous forest litter (DEC), and litter from a ubiquitous emergent macrophyte, *Typha latifolia* (TYP). We focused on emergent macrophytes because they contribute disproportionately to $CH_4$ emissions from lakes and wetlands[21]. We also focused on a single macrophyte species rather than a mixture because they tend to grow in monoculture (e.g., cattail beds), whereas it is more realistic to expect a mix of forest litter inputs (e.g., DEC and CON based on the composition of the littoral forest). The sediments were mixed at 20% OM to approximate the average concentrations found in littoral zones of northern lakes[22], and incubated in laboratory conditions to control other effects, such as temperature, light exposure, and differences in ambient water quality, which confound observational studies. As northern watersheds are

expected to experience a shift in forest composition[23,24] and an increase in emergent macrophyte growth in lakes[25,26], these findings present an additional mechanism to increased mineralization and permafrost thaw[18,27,28] by which climate change can enhance $CH_4$ emission from northern lakes.

## Results and discussion

**Methane production in sediments**. After 150 days of laboratory incubation, $CH_4$ production was over 400-times higher on average from *Typha latifolia* (TYP) sediments than from mixed-coniferous (CON) sediments, almost 2,800-times higher than from mixed-deciduous (DEC) sediments, and 1400-times higher than un-amended controls with 0.3% OM (CTR). In contrast, the CON and DEC treatments did not significantly differ from CTR, suggesting that methanogenesis was inhibited in the sediments amended with forest litter (Fig. 1). Our estimated $CH_4$ production rates for a 150-day growing season ranged from averages of 2.63 mg m$^{-2}$ to $7.22 \times 10^3$ mg m$^{-2}$ amongst the DEC-, CON-, and TYP-amended sediments. These production rates were comparable on a per-area basis to the range and variability of emissions measured in-situ in littoral zones of northern lakes[3], reflecting the close relationship between production and emission in shallow waters[7]. We also found comparable patterns when repeating the experiment with sediments of 10 and 40% OM (Supplementary Fig. 1). A lack of differences in $CO_2$ production rates amongst the amended sediments further suggested that inhibition of methanogenesis and not microbial activity in general was responsible for variation in $CH_4$ production (Supplementary Fig. 2).

**Inhibition of methanogenesis by phenols**. We took two approaches to test the hypothesis that inhibition of

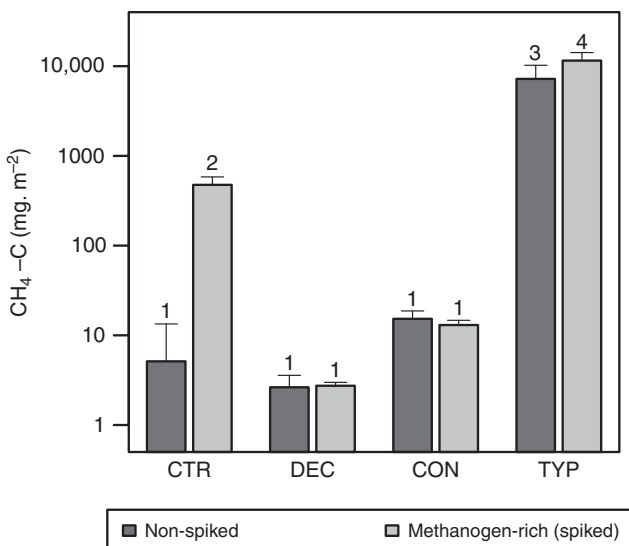

**Fig. 1** $CH_4$ production in amended sediments. Production over a 150-day growing season is orders of magnitude higher in sediments amended with 20% organic matter from emergent macrophyte (*Typha latifolia*; TYP) litter than deciduous (DEC) or coniferous (CON) forest litter. $CH_4$ production increases further with addition of methanogen-rich sediment (i.e., spiked-treatments) only in control (CTR) and TYP sediments. Different numbers (1–4) represent significant differences ($p < 0.05$) among amendments (ANOVA $F_{7, 24} = 39.47$), with $n = 4$ replicates per amendment type. Results are shown on a log scale because of large differences between TYP and the other amendments, and error bars represent standard errors in production estimates

methanogenesis was occurring in the lake sediments amended with forest-derived OM (CON- and DEC-treatments). Firstly, we measured the relative abundance of methanogens using qPCR targeting the *mcrA* gene and found on average $1.72 \times 10^2$ and $1.33 \times 10^4$ fewer *mcrA* copies in the CON and DEC sediments, respectively, compared to the TYP sediments (Fig. 2). These relative abundances mirrored patterns of $CH_4$ production in Fig. 1, suggesting that suppression of methanogen growth was related to decreased production of $CH_4$. Although relative abundance of the *mcrA* gene that we assayed does not entirely equate with specific activity of methanogen communities, there is strong evidence linking it with $CH_4$ production both here (i.e., Fig. 1–2) and in previous studies[29,30]. This link arises because methanogenesis is not known to be a facultative process, but rather the only mechanism methanogens use to generate ATP (e.g., versus facultative denitrifiers in sediments and soils). A large methanogen population would typically be sustained only with concomitant rapid methane production rates.

The second approach we took to test for inhibition of methanogenesis was to conduct a parallel set of incubations where we added a small quantity of a methanogen-rich sediment 'spike' to our treatments at the start of the experiment. Concurrent with our hypothesis of inhibition by plant-derived compounds, there was no change in $CH_4$ production in the DEC or CON sediments with the spike added, but $CH_4$ production doubled in the TYP sediment, and increased most strongly in the un-amended control sediments (Fig. 1). The inhibition of $CH_4$ production in sediments composed of forest-derived compared to macrophyte-derived OM now offers a new mechanism to explain previously described observations in lakes wherein most of the $CH_4$ emissions come from littoral zones covered with macrophytes[3].

Measurements of the biochemical composition of decomposing OM support our conclusion that the inhibition of methanogenesis was caused by phenols from the forest-derived OM. Fluorescence excitation-emission matrices of OM in sediment porewater across all the treatments revealed the presence of a protein-like fluorescence component that was associated with water-soluble phenolic leaf leachates[31,32], in addition to the ubiquitous tryptophan- and tyrosine-like components (Supplementary Fig. 3). Relative concentration of this water-soluble phenol component was lowest in the porewater of the TYP sediments, highest in the DEC sediments, and undetectable in the un-amended CTR sediments. We further found that $CH_4$ production decreased with relative phenol concentration across all the amended sediment types and OM concentrations, suggesting that suppressed methanogenesis in CON and DEC sediments was related to water-soluble phenols (Fig. 3). These phenol leaf leachates were likely inhibiting methanogenesis by reducing enzyme and methanogen activity through direct toxicity[16,33], pH depression, and/or other chemical effects[15,34]. Reduction of methanogenesis can also occur through increased availability of thermodynamically-favorable pathways in sediments (e.g., sulfate reduction), but we did not detect the presence of sulfate reducing bacteria in the sediments (below PCR detection limits; Supplementary Table 2) and so it is likely that sulfate was limiting and/or depleted during the 150-day incubation[33]. Our results complement Freeman et al.[17] who demonstrate that anoxic conditions suppress phenol oxidase, resulting in the buildup of phenols that further inhibit overall decomposition rates (measured as $CO_2$ production). Here we show that the buildup of phenols in anoxic sediments also depends on litter type, and that this has implications to the production of $CH_4$ specifically. We demonstrate that this inhibition is independent of overall rates of decomposition by showing no differences in $CO_2$ production between litter types despite dramatic differences in $CH_4$ production (see Supplementary Fig. 2).

**Implications**. As sediments amended with TYP produced so much more $CH_4$ than forest litter (CON and DEC), our findings

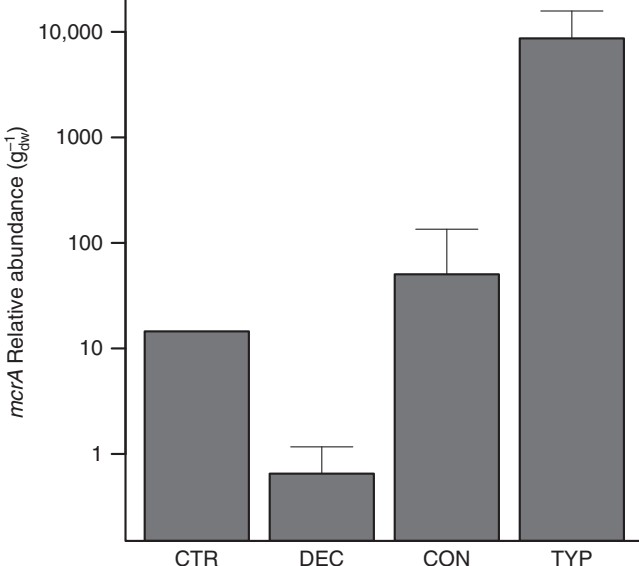

**Fig. 2** Relative abundance of *mcrA* gene copies in amended sediments. Relative abundance is orders of magnitude higher in sediments amended with emergent macrophyte (*Typha latifolia*; TYP) litter than deciduous (DEC) or coniferous (CON) forest litter and mirrors $CH_4$ production in Fig. 1. DNA was pooled across replicates ($n = 4$ per %OM treatment) and expressed as relative abundance per gram dry-weight ($g_{dw}$) of sediment normalized for extraction yield determined by qPCR. Samples were run in triplicate and compared to a standard curve generated from eDNA PCR product to capture the environmental variability in sequences. Error bars for amendments represent standard error across %OM treatments (10, 20, 40%)

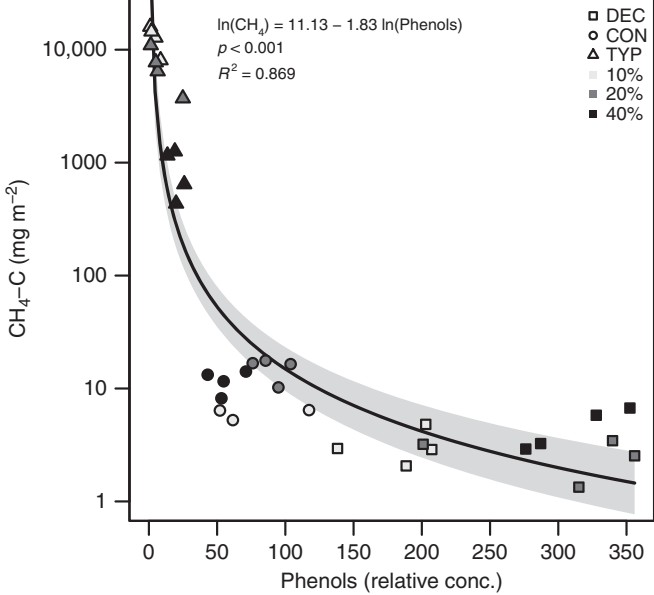

**Fig. 3** $CH_4$ production in sediments declines with phenols. The relationship is shown across OM amendment type (DEC, CON, and TYP) and concentrations (10, 20, 40%), with 95% CI shaded. Concentrations of phenols are relative and determined from fluorescence excitation-emission spectroscopy

may have far-reaching implications for global carbon cycling. For example, species distribution models (SDMs) predict more favorable climatic conditions for the growth of *T. latifolia*—and other emergent macrophytes with similar phenolic foliage content (Supplementary Table 1)—in the Boreal Shield in the coming decades (Supplementary Fig. 4)[26,35]. To consider the implications for $CH_4$ emissions, we overlaid published[24] SDMs produced by Natural Resources Canada for *T. latifolia* onto lakes in the Boreal Shield, an ecozone with relatively homogenous underlying geology and plant communities similar to those in our incubations. By then relating projected occurrence to colonization of suitable lake habitat, we found that the number of Boreal Shield lakes likely to be colonized by *T. latifolia* could double (1.7–2.5 times increase) between 2041 and 2070 (Supplementary Table 3). Assuming no changes other than predicted emergent macrophyte spread, we estimated that the increase in *T. latifolia* alone could elevate $CH_4$ production across Boreal Shield lakes by at least 73% during a 150-day growing season (Supplementary Table 3, Fig. 4). Of course, these estimates are heavily caveated by several assumptions and uncertainties. For example, climate-driven changes in other factors, such as temperature, oxidation potential, and increased forest litterfall production, will certainly influence $CH_4$ production from lake sediments, and all production may not necessarily result in emissions[1]. We have also not accounted for the gas dynamics of living plants, such as rhizosphere processes and aerenchymal transfer that may further enhance emissions where TYP is present[11]. Similarly we have not accounted for the differential mixing of forest-derived OM in sediments resulting from expected shifts in forest composition[36]. However our rough calculation is intended to emphasize that lake sediment chemistry is sufficiently important that it should be considered in earth system models, or at the very least in lake carbon budgets[37].

Methane production in freshwater ecosystems has recently been recognized as an important component of global C cycles[2]. Here we have discovered a new mechanism by which plant-related shifts in sediment chemistry under a changing climate can increase methane production in lakes. This mechanism can account for the observed variability in $CH_4$ emission that has been reported both across and within lakes[2,3], and should enable more precise models and C budgets in northern watersheds.

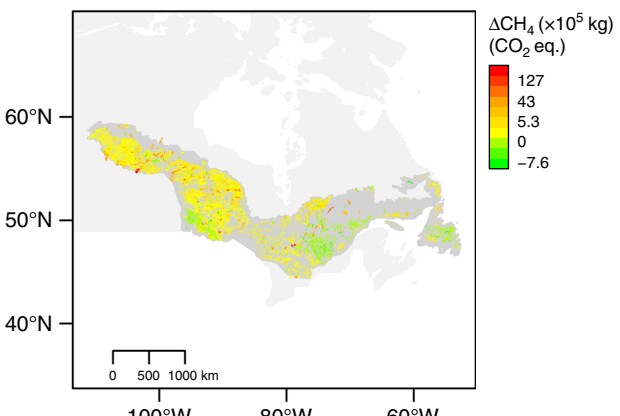

**Fig. 4** Predicted increase in $CH_4$ production across the Boreal Shield. An increase of at least 73% is predicted because of the greater probability of occurrence of *Typha latifolia* alone. Estimated change in production is shown here as change in total kg of $CH_4$ ($CO_2$ equivalents) from current (1971–2001) to future (2041–2070) over a 150-day growing season in each lake under a Composite-AR5 RCP 4.5 climate scenario

## Methods

**Experimental design.** We amended natural sediments with three different sources: senescent coniferous (CON) and deciduous (DEC) litterfall from a transitional/mixed forest stand (Central Ontario: 44°7'22.3"N, 79°30'23.7"W), and senescent *Typha latifolia* (TYP) from Ramsey Lake (in Sudbury, Canada: 46°28'19.8"N 80° 58'19.2"W). TYP is one of several common emergent macrophytes in Boreal Shield lakes, all with similar distribution and phenolic content of foliage (Supplementary Fig. 4, Supplementary Table 1). The CON mix consisted of *Pinus resinosa* and *Pinus strobus*, and the DEC mix consisted primarily of *Acer rubrum*, *Acer saccharum*, *Betula* spp., *Populus tremuloides*, *Ulmus americanum*, *Quercus rubra*, and *Quercus alba*. All OM was oven-dried for 12 h at 60 °C, ground, and sieved to retain only the fine particulate organic matter (FPOM) fraction (≤1 mm).

We mixed the FPOM with a "base inorganic sediment" (0.3% OM, determined by loss-on-ignition at 500 °C for 2 h) to create final OM concentrations (by dry-weight) of 20% across the three amendments (CON, DEC, TYP). We used 20% to approximate typical OM concentrations found in littoral zones of northern lakes[22] (and confirmed in a nearby lake[38]) but we also measured $CH_4$ production with 10 and 40% OM to confirm similarity of patterns across conditions. The base sediment was collected from the shoreline of Geneva Lake (near Sudbury, Canada: 46°45'27.2"N, 81°33'19.8"W) away from *T. latifolia* beds and direct inputs of forest-derived OM and was sieved to exclude particles larger than 2 mm. We distributed the mixed sediments with 250 mL mason jars equipped with rubber septa, with four replicate jars per each %OM and amendment type combination. An estimated 70% of methane production occurs in the top 5 cm of saturated soils[39], so we filled the jars to a depth of 4.5 cm (allowing room for expansion), before saturating them with TOC-scrubbed A10 MilliQ water (EMD Millipore Corp., Darmstadt, Germany). We also created replicated control jars (CTR) containing only base sediment, otherwise constructed and treated in the same manner.

We duplicated the 20% OM experimental setup with a "methane-rich spike". The spike consisted of replacing 5% of the base sediment with sediment from the top 5 cm of a littoral site in Ramsey Lake previously known to have high rates of methane production. Amendments of CON, DEC, TYP were adjusted for the 2.8% OM content of the spike sediment to ensure final OM concentrations of 20% (dry-weight).

**$CH_4$ and $CO_2$ production.** We incubated the sediments and periodically collected headspace samples to measure $CH_4$ and $CO_2$ production after 150 days, representative of the length of a growing season in the Boreal Shield. The sediments were incubated in a BioChambers SPC-56 growth chamber in the dark at 20.5 °C. At the start of the incubations, headspace air in each jar was replaced four times with $N_2$ using a vacuum manifold to ensure anaerobic conditions and removal of atmospheric $CO_2$ and $CH_4$. We collected headspace gas by homogenizing 10 mL of $N_2$ into headspace prior to extracting a 10 mL gas sample by syringe, repeating this periodically over the 150-day incubation to ensure we reached a plateau $CO_2$ and $CH_4$ production in all sediments (Supplementary Fig. 5). The total volume removed was quantified and used to correct headspace volume throughout the incubation. Both $CH_4$ and $CO_2$ were detected as $CH_4$ using a SRI 8610C gas chromatograph (0.5 mL sample loop, 105 °C column temperature), and production was calculated at the end of 150 days, adding back the portions that were removed and expressing totals as mg m$^{-2}$ of dry sediment given an area of 28.3 cm$^2$ for each jar.

**Relative phenol concentration.** To measure relative phenol concentration, we collected porewater from each jar after the 150-day incubations and filtered the samples through 0.5 μm glass fiber filters. Samples were acidified to pH < 2 with HCl, and stored in airtight vials at ~4 °C. Fluorescence EEMs (excitation–emission matrices) were generated using an Agilent Cary Eclipse Fluorescence Spectrophotometer with a 1 cm path-length cuvette. EEMs were generated from excitation and emission intensities (EX: 250–450 nm in 5 nm steps, EM: 300–600 nm in 2 nm steps) that were adjusted for inner-filter effects with absorbance as measured with an Agilent Cary 60 UV-Vis Spectrophotometer. All EEM sample correction and PARAFAC modeling was done in Matlab R2015b according to the methods outlined in ref. 40. Five PARAFAC components were validated by a split-half method[40], explaining 98.7% of the variation in the EEMs. Components C1 and C2 were comparable to common humic-like components, with maximum excitation/emission intensities of (310/414 nm) and (345/462 nm) respectively. C3 was similar to the common tryptophan protein-like component (280/354 nm), and C4 the common tyrosine protein-like component (270/306 nm). Component C5 (275/318 nm) was identified as a protein-like component that is associated with leaf litter phenol leachates[31,32] (Supplementary Fig. 3).

We further confirmed the association between C5 and litter-derived phenol leachates using ultra-high resolution mass spectrometry data collected on a subset of samples in our PARAFAC model from an accompanying field-scale incubation study broadly described in ref. 38 (Supplementary Fig. 6). We extracted DOM from 5.5 mL of filtered and acidified (HCl, pH 2) porewater from sediment in each of 39 field-deployed mesocosms using styrene divinyl benzene polymer solid phase extraction (SPE) cartridges (Agilent Bond Elut PPL, 100 mg)[41]. The methanol extract was stored at −20 °C in the dark until further analysis. SPE-DOC concentration was determined by drying an aliquot of the SPE-DOM extract (at 40 °C) and re-dissolving it in ultrapure water. The methanol extracts were diluted to yield a DOC concentration of 5 mg L$^{-1}$ in ultrapure water and MS grade methanol

and analyzed with Fourier-transform ion cyclotron resonance mass spectrometry (FT-ICR-MS) on a 15 Tesla solariX (Bruker Daltonik, Bremen, Germany) at the University of Oldenburg using electrospray ionization in negative mode with 4 kV capillary voltage. Data were acquired in broadband mode using 8 megaword data sets and a range of 92–2000 Da with 125 scans accumulated per mass spectrum. Mass spectra were calibrated internally with a list of known compounds in the targeted mass range (achieved mass accuracy < 0.1 ppm). Molecular formulae were then assigned with the following restrictions: $C_{1-130}H_{1-200}O_{1-50}N_{0-4}S_{0-2}P_{0-1}$ to masses above the method detection limit[42]. Additionally, masses detected in less than two samples were removed prior to further analysis. Signal intensities of assigned peaks were normalized to the sum of all peak intensities with identified molecular formulae in each sample.

Relative phenol leachate concentration was estimated as the product of proportional C5 fluorescence and total dissolved organic carbon (DOC) concentration in sediment porewater, as measured on a Shimadzu TOC-5000A in FPOC mode.

**Methanogen suppression**. To compare the relative abundance of methanogens between samples, DNA was first extracted in duplicate using the MoBio PowerSoil kit (MoBio, Carlsbad, CA, USA). qPCR was then carried out in triplicate on pooled DNA extractions to better characterize the communities from the 4 sediment replicates of the CTR and each OM type mixed at 10, 20 and 40% concentration. The *mcrA* gene was targeted using mlasF (5′-GGYGGTGTMGGDTTCACM-CARTA-3′) and mcrA-rev (5′-CGTTCATBGCGTAGTTVGGRTAGT-3′) primers as in ref. [29]. Reaction conditions were: a 5-min initial denaturation at 95 °C, followed by 45 cycles of 95 °C for 15 s, 55 °C for 30 s, and 72 °C for 30 s. Then a final denaturation for 1 min at 95 °C, 30 s at 42 °C and 95 °C for 30 s. The qPCR was done using Biorad's iTaq Universal SYBR Green Supermix on an Agilent Technologies Stratagene Mx3005P at Laurentian University. A standard curve was generated by serially diluting an extracted band from amplified eDNA and run in triplicate along with the samples generating an $R^2 = 0.999$ and efficiency of 97.2%. Dissociation curves indicated a pure product, which was confirmed on a 1.5% agarose gel. eDNA was quantified and purity was checked (260/280 nm ratio) using a Take3 spectrophotometry system on a Synergy HI microplate reader (BioTek, Winooski VT, USA). Dissociation curves indicated a pure product, which was confirmed on a 1.5% agarose gel. The results were calculated by averaging the triplicate Ct values, and abundances were standardized relative to the control and expressed per dry weight of sediment normalized for extraction efficiency.

Suppression of methanogens could also be caused by sulfate reducing bacteria (SRB). To test for this, we used PCR to target the SRB-specific *dsrA* gene and 16S rRNA gene deep sequencing to evaluate their abundance. DNA was extracted from 0.5 g of soil in duplicate using the MoBio PowerSoil kit (Mo Bio, Carlsbad, CA, U.S.A.). Samples were then pooled to provide two replicates of the control (CTR) and two replicates per %OM treatment (10, 20, 40%) for a total of 6 samples for each sediment type (DEC, CON, TYP). Samples were sequenced on an Illumina MiSeq using the prokaryote primers Pro 341 F (5′-CCTACGGGNBGCASCAG-3′) and Pro805R (5′-GACTACNVGGGTATCTAATCC-3′)[43] by Metagenome Bio Inc. Resulting sequences were merged using Pandaseq and further quality filtered and taxonomy was assigned from the Green Genes database using Usearch v8.1.1861 and QIIME, respectively[44–46]. Data were analyzed in R with the Phyloseq package[47], and abundance data were corrected using the DeSEq2 package in R[48,49]. PCR was used on representative samples using the *dsrA* primers DSR1-F (5′-ACSCACTGGAAGCACGGCGG-3′) and DSR-R (5′-GTGGMRCCGTGCAK RTTGG-3′)[50]. The PCR was run using Phire Hot Start II PCR Mastermix (ThermoFisher Scientific), conditions were an initial denaturation at 98 °C for 2 min and 30 cycles of 98 °C for 20 s, 59 °C for 20 s, 72 °C for 40 s and a final extension for 2 min at 72 °C.

**Ecosystem-scale emissions**. We estimated the impact of increased *T. latifolia* occurrence on $CH_4$ production during the growing season by applying our estimated production rates (mg m$^{-2}$) to current and projected aerial cover (m$^2$) for Boreal Shield lakes. Surface areas were obtained from the Global Lakes and Wetlands Database (GLWD)[8] for waterbodies between 0.1 and 1000 km$^2$ in size and located within the Boreal Shield (spatially delineated by the National Ecological Framework for Canada[51] as an area of 1.8 million km$^2$ located between ca. 45°N and 60°N characterized by underlying Precambrian bedrock). For each lake, we extracted current and projected probability of occurrence of *T. latifolia* from Natural Resources Canada MaxEnt species distribution model raster data, which was developed by combining species occurrence data with actual climate data (for current estimates) and with climate models (GCMs)[24,35]. We used projected MaxEnt occurrences for the timeframe of 2041–2070 incorporating uncertainty by using five climate models [canESM2, hadGEM2-ES, CESM1(CAM5), MIROC-ESM-CHEM, and composite-AR5] each with three future emission scenarios (RCP 2.6, 4.5 and 8.5).

We then used the range in the probability of occurrence data to estimate a range in projected suitable habitat and thus proportional coverage within Boreal Shield lakes. Suitable emergent macrophyte habitat (shallow littoral) areas are not widely available, so we used published regressions indicating a maximum of 28% of lake area to be covered by emergent macrophytes, on average, for Boreal lakes in our size range[52]. We then estimated coverage as the product of the probability of occurrence of TYP and 28% of the total lake areas that were widely available across

the Boreal from the GLWD. Thus, a probability of occurrence of 1 meant all suitable habitat, or 28% of total lake area, was likely to be covered by TYP in a given lake. Realistically some of this habitat would be occupied by other emergent macrophytes with similar habitat preferences, but the most common species have a similar distribution and comparable foliage phenolic content to TYP (Supplementary Fig. 4, Supplementary Table 1). We then calculated total $CH_4$ production as the product of the rate of production (mg m$^{-2}$) in our incubation study and coverage by TYP (m$^2$, current and projected), propagating uncertainty from climate models and scenarios along with variation in our $CH_4$ production estimates. Estimates were scaled up to 100% of the sediment profile assuming our 5 cm surficial sediments represented 70% of total production[39] and presented in $CO_2$ equivalents (1 kg $CH_4$ = 25 kg $CO_2$) to maintain consistency with global emission estimates in ref. [2].

**Statistical analysis**. To compare production rates across OM type, we performed a one-way ANOVA. The ANOVA included the effect of type and its interaction with the methanogen spike with the baseline (intercept) group adjusted to compare significance among groups. The ANOVA was repeated for 10, 20, and 40% OM separately. We then fit a log–log model to test for an effect of relative phenol concentrations on CH4 production. All analyses were done in R v. 3.3.0[49].

**Data availability**. All sequence data have been deposited in the NCBI Sequence Read Archive under BioProject accession code PRJNA347436. All other datasets generated during and/or analyzed during the current study are available from the corresponding author on reasonable request.

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

## Acknowledgements

We acknowledge the support of staff and colleagues at the Living with Lakes Centre, Laurentian University, in Sudbury Canada. We also thank John Pedlar and Dan McKenney at Natural Resources Canada for providing spatial MaxENT data, and Michael Seidel, Ina Ulber, and Katrin Klaproth at ICBM for help with FT-ICR-MS analysis and interpretation. We thank three anonymous reviewers for improving an earlier draft of this manuscript. Funding was provided by NERC Standard Grant NE/L006561/1 to AJT.

## Author contributions

E.J.S.E., M.A.C., J.M.G., N.C.S.M., N.B., and A.J.T. conceived the study. E.J.S.E., M.A.C., and K.M.Y. collected the data, H.O. and T.D. assisted with analysis and interpretation of FT-ICR-MS data, and E.J.S.E. analyzed the data and wrote the manuscript with input from all authors

## Additional information

**Competing interests:** The authors declare no competing interests.

