## [Peer review file · Nature Communications]

Editorial Note: The figure on page 4 of this Peer Review File is reproduced with permission from GBIF (Typha latifolia L. in GBIF Secretariat (2017). GBIF Backbone Taxonomy. Checklist Dataset <https://doi.org/10.15468/39omei> accessed via GBIF.org on 2018-04-06).

Reviewers' comments:

Reviewer #1 (Remarks to the Author):

This study is an excellent, well written, and thoroughly convincing laboratory study which shows that the litter from a common wetland plant (*Typha latifolia*) is a far better substrate for methanogenesis than is the litter from mixed conifers or deciduous trees. In fact, when the litter from *Typha* was added to sediments from a lake, 400 times more methane was produced than when the forest tree litter was added. While the actual study is excellent, I find the connection to global change is based on very little. The authors argue, citing two fairly obscure papers, that climate change will result in shifts of aquatic plants. *Typha latifolia*, strictly speaking, is a wetland plant and not the sort of macrophyte being discussed in the main paper used as support for this argument (Alahuhta et al. 2011). In fact, as far as I know *Typha latifolia* does not occur in Boreal Region of Scandinavia, upon which Alahuhta et al. is based). I think the paper needs two things to make is a Nature Communications paper. 1) the litter of at least several aquatic macrophytes needs to be compared to forest litter and each other; and 2) the authors needs to present much stronger evidence that climate change will actually increase the kind of macrophytes that support high rates of methanogenesis. Of course, if the authors were to drop the claims about climate change, that would help with the paper but might make it less appealing a Nature Comm.

Reviewer #2 (Remarks to the Author):

Comments to

"Climate-driven shifts in sediment chemistry enhance methane production in northern lakes"
by Emilson et al.

The present global estimates on freshwater CH₄ emission are still highly uncertain due to limited geographical coverage of data available and limited knowledge on the role of diffusive flux, ebullition and plant mediated transport included in CH₄ emissions. In order to decrease the uncertainty in methane production in northern lakes the authors studied the role of polyphenols in determining CH₄ production in lake sediments and concluded that earth system models and carbon budgets should consider the effects of plant communities on sediment chemistry and ultimately CH₄ emissions at a global scale.

The work is convincing and the results are interesting. However, it would be easier for the reader to pick up the results from this study if some conclusions would be modified to better reflect the new results, i.e. the strong correlation between polyphenols and CH₄ production (Figure 4).

Abstract

"Climate change is predicted to shift plant community composition, and thus change the quality of inputs into detrital food webs, with the potential to affect CH₄ production."

"Our results now suggest that earth system models and carbon budgets should consider the effects of plant communities on sediment chemistry and ultimately CH₄ emissions at a global scale."

- Current earth system models are still very simplified descriptions on all important drivers included in methane production.
- Sediment chemistry is definitely important, but not the only missing piece. Physical component of earth system models is becoming increasingly comprehensive, but the biogeochemical components suffer from a lack of comprehensive global-scale observational data (WIREs Clim Change 2011).

Introduction

The authors conclude that plant-related CH₄ fluxes remain one of the least-understood components of the global methane budget (Bastviken et al. 2011).

- I agree. Nevertheless, different plant species produce variable amounts of CH₄ which can result

in large differences in regional CH₄ production estimates if only total vegetation coverage is used as an estimate for CH₄ production. Species specific CH₄ emission rates and areal coverage of the dominating species contributing to CH₄ production should be considered when estimating the total regional emissions of CH₄ (Bergström et al. 2007).

Results and Discussion

“Relative concentration of this water-soluble polyphenol component was lowest in the pore water of the TYP sediments, highest in the DEC sediments, and undetectable in the un-amended CTR sediments.”

- why to use only TYP sediments to represent macrophytes, when the CON mix consisted of *Pinus resinosa* and *Pinus strobus*, and the DEC mix consisted primarily of *Acer rubrum*, *Acer saccharum*, *Betula* spp., *Populus tremuloides*, *Ulmus americanum*, *Quercus rubra*, and *Quercus alba*?
- why is water-soluble polyphenol component highest in the DEC sediments? - is this a generally known pattern in northern lakes?

“Here we have discovered a new mechanism by which plant-related shifts in sediment chemistry under a changing climate can increase methane production in lakes. This mechanism can account for the observed variability in CH₄ emission that has been reported both across and within lakes (Carmichael et al. 2014, Juutinen et al. 2003) and should enable more precise models and C budgets in northern watersheds.”

- similar kind of mechanism was suggested to regulate carbon sequestration in peatlands (Freeman et al. 2001).

Materials and methods

“Suitable emergent macrophyte habitat (shallow littoral) areas are not widely available, so we used published regressions indicating a maximum of 28% of lake area to be covered by emergent macrophytes, on average, for Boreal lakes in our size range (Duarte et al. 1986). We then estimated coverage as the product of the probability of occurrence and 28% of the total lake areas that were widely available across the Boreal from the GLWD. Thus, a probability of occurrence of meant all suitable habitat, or 28% of total lake area, was covered in a given lake. We then calculated total CH₄ production as the product of the rate of production (mg m⁻²) in our incubation study and coverage by *T. latifolia* (m², current and projected), propagating uncertainty from climate models and scenarios along with variation in our CH₄ production estimates.”

- *Typha latifolia* is not the only plant in lake littorals, i.e. vegetation cover does not just reflect the coverage of *Typha latifolia*.
- Large differences in CH₄ emission have been shown to occur among different macrophytes. Total macrophyte coverage was not the key driver, the species specific CH₄ fluxes varied significantly. The emissions from the stands of floating-leaved species were negligible compared to the emissions from stands of *P. australis* and *E. fluviatile*. *Typha latifolia* played a very minor role in regional CH₄ emissions in boreal lakes (Bergström et al. 2007).

Bastviken, D., Tranvik, L. J., Downing, J. A., Crill, P. M. & Enrich-Prast, A. Freshwater Methane Emissions Offset the Continental Carbon Sink. *Science*, 331, 50–50 (2011).

Bergström, I., Mäkelä, S., Kankaala, P. & Kortelainen, P. 2007. Methane efflux from littoral vegetation stands of southern boreal lakes: and upscaled, regional estimate. *Atmospheric Environment* 41: 339-351.

Carmichael, M. J., Bernhardt, E. S., Bräuer, S. L. & Smith, W. K. The role of vegetation in methane flux to the atmosphere: Should vegetation be included as a distinct category in the global methane budget? *Biogeochemistry* 119, 1–24 (2014).

Duarte, C. M., 1. Kalff, and R. H. Peters. 1986. Patterns in biomass and cover of aquatic macrophytes in lakes. *CJFAS* 43: 1900-1908.

Freeman, C., Ostle, N. & Kang, H. An enzymic 'latch' on a global carbon store. *Nature* 409, 149–150 (2001).

Juutinen, S. et al. Major implication of the littoral zone for methane release from boreal lakes. *Global Biogeochem. Cycles* 17, 1–11 (2003).

WIREs *Clim Change* 2011, 2:783–800. doi: 10.1002/wcc.148

Reviewer #3 (Remarks to the Author):

The manuscript NCOMMS-17-22360 describes the effect of polyphenols from an extract of senescent coniferous (CON), deciduous (DEC) and senescent *Typha latifolia* (TYP) on the methanogenesis in freshwater water sediments. This lab-experiment was up-scaled by a modelling approach to estimate its effect on a larger scale.

The story is of high interest of the readers and the data set is almost sound. The main problem I see in this study is, that the author think that the main effect on methanogenesis is been caused by the addition of polyphenols to their incubations. From my point of view polyphenols are a major component in the extracts from CON, DEC and TYP but by far not the only component. Therefore, the authors should make a detailed analyses which other components (beside polyphenols) they may have also co-extracted and discuss this other components in detail. Other option would be to purify polyphenols from these extracts and use those more "defined" compounds as effector for their study. In the latter, the extraction method may modify natural other compounds and may also shift their effect on methanogenesis.

In addition, I found some more open tasks to be improved in this manuscript:

Introduction

I.94 mcrA gene copies

Why have the authors conducted analyses on DNA-level? DNA-level is just related to presence and may proliferation of methanogens but not for their activity. Much better would be to use mcrA transcripts or proteome analyses to estimate the activity.

Material & Methods

First, please include the description of your control (CTR treatment) in your Experimental design.

I.190 Please also show the CH₄ and CO₂ data from all incubation times! Is the rate of CH₄ emission also affected by the amended CON, DEC or TYP?

I.209-213 I know that the analyses of polyphenols is not easy, but have the amended CON, DEC or TYP differed to each other in their composition and concentration of polyphenol? The composition and concentration of polyphenol should be different and the author should state that differences in the initial source of CON, DEC and TYP both before and after addition.

I.240/I.282/I.4 of supplement: 16S rRNA gene!

Figure 1: Please change spiked with addition of methanogen-rich sediment. Indicate the number of replicates.

Figure 2: Why do the authors do not present absolute mcrA gene copy numbers per gram dw? Relative abundances are not meaningful here. Pooling of DNA is not helpful as now only technical replicates are shown. Line 438-439 is not clear for me. Where do the error bars come from?

Supplementary methods

I.6 The authors should not pool sample to do statistics with biological replicates!

Supplementary Figure1: The authors should include the data of addition of methanogen-rich sediment.

We have indicated our responses in blue, and changes to manuscript text are shown in red. Changes in the revised manuscript and revised SI are highlighted in yellow. Line numbers in responses refer to line numbers in the revised document.

Reviewers' comments:

Reviewer #1 (Remarks to the Author):

This study is an excellent, well written, and thoroughly convincing laboratory study which shows that the litter from a common wetland plant (*Typha latifolia*) is a far better substrate for methanogenesis than is the litter from mixed conifers or deciduous trees. In fact, when the litter from *Typha* was added to sediments from a lake, 400 times more methane was produced than when the forest tree litter was added. While the actual study is excellent, I find the connection to global change is based on very little. The authors argue, citing two fairly obscure papers, that climate change will result in shifts of aquatic plants. *Typha latifolia*, strictly speaking, is a wetland plant and not the sort of macrophyte being discussed in the main paper used as support for this argument (Alahuhta et al. 2011). In fact, as far as I know *Typha latifolia* does not occur in Boreal Region of Scandinavia, upon which Alahuhta et al. is based).

We thank the Reviewer for recognizing the strength of our study. We have substantially strengthened the connection to global change and have expanded on this below and with revisions in the manuscript where specified. While it is true that *T. latifolia* is a common wetland plant, it is also a very common emergent macrophyte in nearshore ("wetland") areas of northern lakes (see example in Finnish Boreal lakes: Toivonen and Huttunen 1995 *Aquatic Botany*), and is widely distributed in both North America and the Boreal Region of Scandinavia. A simple search on GBIF confirms this with >100k records of occurrence that are heavily concentrated in Scandinavia (<https://www.gbif.org/species/5289423>):

I think the paper needs two things to make is a Nature Communications paper.

1) the litter of at least several aquatic macrophytes needs to be compared to forest litter and each other;

We have now added Table S3 where we synthesise 12 published studies to compare the litter of several aquatic macrophytes with forest litter, and mapped the projected changes of these macrophytes in Figure S4. We have also added the following sentences to the Introduction and Methods to clarify our reasoning for selecting *T. latifolia*:

Lines 71-75: *“We focused on emergent macrophytes because they contribute disproportionately to CH₄ emissions from lakes and wetlands¹⁹. We also focused on a single macrophyte species rather than a mixture because they tend to grow in monoculture (e.g.: cattail beds), whereas it is more realistic to expect a mix of forest litter inputs (e.g.: DEC and CON based on the composition of the littoral forest).”*

Lines 178-180: *“TYP is one of several common emergent macrophytes in Boreal Shield lakes, all with similar distribution and phenolic content of foliage (Supplementary Fig. S4, Table S3).”*

2) the authors needs to present much stronger evidence that climate change will actually increase the kind of macrophytes that support high rates of methanogenesis. Of course, if the authors were to drop the claims about climate change, that would help with the paper but might make it less appealing a Nature Comm.

As explained above, we have found that macrophytes in general have litter qualities that can support high rates of methanogenesis, and presented new evidence that climate change will increase their distributions. We used species distribution models (SDMs) generated by Natural Resources Canada with an exceptionally large observational dataset covering North America (McKenney et al. 2007). These SDMs support our contention that suitable habitat for several emergent macrophytes (including *T. latifolia*) will shift northward, increasing their probability of occurrence in the Boreal Shield (see Supplementary Fig. S4). These species also produce extremely high numbers of seeds and have high dispersal capabilities (Grace and Harrison 1986, *Can. J. Plant Sci.*; Soons and Ozinga 2005, *Diversity and Distributions*), making northwards migration very likely. We have compiled all available published data on foliage phenolic content (see Supplementary Table S3) to show that all of these emergent macrophytes have comparable phenolic content and thus similar potential to affect CH₄ production. Note that we have also incorporated much of the uncertainty in these estimates by including 5 different climate change models each with 3 emission scenarios within our range of expected outcome (see lines 277-280).

We now refer to the added Figure S4 and Table S3 in the Discussion (and the Introduction and Methods, as mentioned above):

Lines 145-149: *“For example, species distribution models (SDMs) predict more favourable climatic conditions for the growth of *T. latifolia*—and other emergent macrophytes with similar foliar phenolic content (Supplementary Table S3)—in the boreal Shield in the coming decades (Supplementary Fig. S4)*

23,32 ”

Reviewer #2 (Remarks to the Author):

The present global estimates on freshwater CH₄ emission are still highly uncertain due to limited geographical coverage of data available and limited knowledge on the role of diffusive flux, ebullition and plant mediated transport included in CH₄ emissions. In order to decrease the uncertainty in methane production in northern lakes the authors studied the role of polyphenols in determining CH₄ production in lake sediments and concluded that earth system models and carbon budgets should consider the effects of plant communities on sediment chemistry and ultimately CH₄ emissions at a global scale.

The work is convincing and the results are interesting. However, it would be easier for the reader to pick up the results from this study if some conclusions would be modified to better reflect the new results, i.e. the strong correlation between polyphenols and CH₄ production (Figure 4).

We thank the Reviewer for recognizing the importance of our results. We have addressed specific comments below, including in ways that better reflect the novelty of our results.

Abstract

“Climate change is predicted to shift plant community composition, and thus change the quality of inputs into detrital food webs, with the potential to affect CH₄ production.”

“Our results now suggest that earth system models and carbon budgets should consider the effects of plant communities on sediment chemistry and ultimately CH₄ emissions at a global scale.”

- Current earth system models are still very simplified descriptions on all important drivers included in methane production.

- Sediment chemistry is definitely important, but not the only missing piece. Physical component of earth system models is becoming increasingly comprehensive, but the biogeochemical components suffer from a lack of comprehensive global-scale observational data (WIREs Clim Change 2011).

We agree that earth system models are very simplified and suffer from a lack of global biogeochemical data. Here we present a link between vegetation distributions (which can be globally mapped) and biogeochemical processes as one novel process to enhance these models. We have further explained this point by adding text to the Introduction (provided below in response to the next comment) and adding text to the Discussion (provided below in responses to the point about *Freeman et al. (2001)* added to lines 137-143).

Introduction

The authors conclude that plant-related CH₄ fluxes remain one of the least-understood components of the global methane budget (Bastviken et al. 2011).

- I agree. Nevertheless, different plant species produce variable amounts of CH₄ which can result in large differences in regional CH₄ production estimates if only total vegetation coverage is used as an estimate for CH₄ production. Species specific CH₄ emission rates and areal coverage of the dominating species contributing to CH₄ production should be considered when estimating the total regional emissions of CH₄ (Bergström et al. 2007).

We agree that aerial coverage of dominating species should be considered when budgeting total emissions from a given lake, but our goal was instead to estimate changes in regional production that

would occur from a single mechanism in isolation: increases in emergent macrophytes, which contribute disproportionately to CH₄ emissions from lakes (See ref 19 in manuscript). Nonetheless, we have provided more detail for why we focused on this specific macrophyte in response to the next comment, and have made the following addition to the Introduction:

Lines 50-55: *“Another explanation is that the activity of sediment microbial communities is inhibited, to varying degrees, by the breakdown of different OM sources¹⁴, resulting in variation in the production of CH₄ in littoral sediments. Therefore, regional estimates of CH₄ emissions may need to consider the aerial coverage of different plant species and functional types if they contribute OM that differentially influence rates of sediment CH₄ production.”*

Results and Discussion

“Relative concentration of this water-soluble polyphenol component was lowest in the pore water of the TYP sediments, highest in the DEC sediments, and undetectable in the un-amended CTR sediments.”

- why to use only TYP sediments to represent macrophytes, when the CON mix consisted of *Pinus resinosa* and *Pinus strobus*, and the DEC mix consisted primarily of *Acer rubrum*, *Acer saccharum*, *Betula* spp., *Populus tremuloides*, *Ulmus americanum*, *Quercus rubra*, and *Quercus alba*?

We have added the following sentences to clarify our reasoning for the selection of *T. latifolia* specifically within the Introduction and Methods, and added Figure S4 and table S3 to support our argument.

Lines 71-75: *“We focused on emergent macrophytes because they contribute disproportionately to CH₄ emissions from lakes and wetlands¹⁹. We also focused on a single macrophyte species rather than a mixture because they tend to grow in monoculture (e.g.: cattail beds), whereas it is more realistic to expect a mix of forest litter inputs (e.g.: DEC and CON based on the composition of the littoral forest).”*

Lines 178-180: *“TYP is one of several common emergent macrophytes in Boreal Shield lakes, all with similar distribution and phenolic content of foliage (Supplementary Fig. S4, Table S3).”*

- why is water-soluble polyphenol component highest in the DEC sediments? - is this a generally known pattern in northern lakes?

There are no studies that we are aware of that measure soluble phenols from deciduous litter-derived sediments, which makes our work all the more important. Nonetheless, we have compiled total phenolic content of leaf litter from common northern macrophytes and trees into a new Table S3. Little total phenol data are generally available for coniferous foliage and methods are not consistent. Regardless, we find overlap in reported phenolic content between CON and DEC litter, but with values consistently higher than that of TYP and other emergent macrophytes.

“Here we have discovered a new mechanism by which plant-related shifts in sediment chemistry under a changing climate can increase methane production in lakes. This mechanism can account for the observed variability in CH₄ emission that has been reported both across and within lakes (Carmichael et al. 2014, Juutinen et al. 2003) and should enable more precise models and C budgets in northern

watersheds.”

- similar kind of mechanism was suggested to regulate carbon sequestration in peatlands (Freeman et al. 2001).

Freeman et al. (2001) are discussing a particular process by which the slow rates of decomposition observed in peatlands are a result of the suppression of enzymes by phenols, because of the inhibition of phenol oxidase activity in anoxic conditions. We have added the following to the Discussion to better illustrate how our results complement those of Freeman et al. (2001):

Lines 137-143: “Our results complement Freeman et al.¹⁷ who demonstrate that anoxic conditions suppress phenol oxidase, resulting in the buildup of phenols that inhibit overall decomposition rates (measured as CO₂ production). Here we show that the buildup of phenolic compounds in anoxic sediments also depends on litter type, and that this has implications to the production of CH₄ specifically. We demonstrate that this inhibition is independent of overall rates of decomposition by showing no differences in CO₂ production between litter types despite dramatic differences in CH₄ production (see Supplementary Fig. S2).”

Materials and methods

“Suitable emergent macrophyte habitat (shallow littoral) areas are not widely available, so we used published regressions indicating a maximum of 28% of lake area to be covered by emergent macrophytes, on average, for Boreal lakes in our size range (Duarte et al. 1986). We then estimated coverage as the product of the probability of occurrence and 28% of the total lake areas that were widely available across the Boreal from the GLWD. Thus, a probability of occurrence of 1 meant all suitable habitat, or 28% of total lake area, was covered in a given lake. We then calculated total CH₄ production as the product of the rate of production (mg m⁻²) in our incubation study and coverage by *T. latifolia* (m², current and projected), propagating uncertainty from climate models and scenarios along with variation in our CH₄ production estimates.”

- *Typha latifolia* is not the only plant in lake littorals, i.e. vegetation cover does not just reflect the coverage of *Typha latifolia*.

We scaled the vegetation cover (i.e: 28% of lake area) by the probability of occurrence of *T. latifolia* (current and projected, from SDMs discussed on lines 273-279). It is true that some of this littoral area would be colonized by other emergent macrophytes with similar habitat preference, but these also have similar phenolic content and likely to exhibit similar effects on CH₄ production. We have clarified this with two additions to the Methods and the addition of Supplementary Fig. S4 and Table S3:

Lines 178-180: “TYP is one of several common emergent macrophytes in Boreal Shield lakes, all with similar distribution and phenolic content of foliage (Supplementary Fig. S4, Table S3).”

Lines 285-291: “We then estimated coverage as the product of the probability of occurrence of TYP and 28% of the total lake areas that were widely available across the Boreal from the GLWD. Thus, a probability of occurrence of 1 meant all suitable habitat, or 28% of total lake area, was likely to be covered by TYP in a given lake. Realistically some of this habitat would be occupied by other emergent macrophytes, but the most common species have a similar distribution and comparable foliage phenolic content to TYP (Supplementary Fig. S4, Table S3).”

- Large differences in CH₄ emission have been shown to occur among different macrophytes. Total macrophyte coverage was not the key driver, the species specific CH₄ fluxes varied significantly. The emissions from the stands of floating-leaved species were negligible compared to the emissions from stands of *P. australis* and *E. fluviatile*. *Typha Latifolia* played a very minor role in regional CH₄ emissions in boreal lakes (Bergström et al. 2007).

The observation that emergent macrophyte stands such as *E. fluviatile* and *P. australis* emit considerably more CH₄ than floating-leaved species is exactly why we chose to focus on this functional group. The lakes in Bergström et al. 2007 were dominated by stands of *E. fluviatile* and *P. australis*. Although *T. latifolia* was present at lower coverage, Bergström et al. 2007 did not estimate its associated CH₄ fluxes. By contrast, lakes in the Boreal Shield region that we focus upon are much more dominated by *T. latifolia*. We have shown this pattern in Supplementary Fig. S4 and clarified this with the following change to the Introduction and Methods:

Lines 71-75: *"We focused on emergent macrophytes because they contribute disproportionately to CH₄ emissions from lakes and wetlands¹⁹. We also focused on a single macrophyte species rather than a mixture because they tend to grow in monoculture (e.g.: cattail beds), whereas it is more realistic to expect a mix of forest litter inputs (e.g.: DEC and CON based on the composition of the littoral forest)."*

Lines 178-180: *"TYP is one of several common emergent macrophytes in Boreal Shield lakes, all with similar distribution and phenolic content of foliage (Supplementary Fig. S4, Table S3)."*

We have also added text on line 50-55, as explained above, to explain that while we have provided one specific example, regional estimates of CH₄ emissions may need to consider the aerial coverage of different plant species and functional types.

Reviewer #3 (Remarks to the Author):

The manuscript NCOMMS-17-22360 describes the effect of polyphenols from an extract of senescent coniferous (CON), deciduous (DEC) and senescent *Typha latifolia* (TYP) on the methanogenesis in freshwater water sediments. This lab-experiment was up-scaled by a modelling approach to estimate its effect on a larger scale.

The story is of high interest of the readers and the data set is almost sound. The main problem I see in this study is, that the author think that the main effect on methanogenesis is been caused by the addition of polyphenols to their incubations. From my point of view polyphenols are a major component in the extracts from CON, DEC and TYP but by far not the only component. Therefore, the authors should make a detailed analyses which other components (beside polyphenols) they may have also co-extracted and discuss this other components in detail. Other option would be to purify polyphenols from these extracts and use those more "defined" compounds as effector for their study. In the latter, the extraction method may modify natural other compounds and may also shift their effect on methanogenesis.

We thank the Reviewer for recognizing the importance of our study and soundness of our dataset. As they have recommended, we have carried out a detailed analysis of what other components we have extracted to complement our interpretation of the phenolic fluorescence PARAFAC component (C5). Our analysis of ultra-high resolution mass spectrometry data collected on a subset of samples in our PARAFAC model showed that several molecules exclusively correlated to C5 were unsaturated and less oxidised compounds of low O/C ratio, typically classified as vascular plant-derived polyphenols, phenolic compounds, and other highly condensed aromatic compounds (Seidel et al. 2017 *Front Earth Sci*; Kellerman et al. 2014 *Nat Commun*). There were also several aliphatic compounds exclusively correlated to C5 with low O/C ratio, as might be expected if the increasing availability of C5 inhibited methanogenesis and maintained fresh leaf leachate compounds in solution. We agree with the Reviewer that this approach is better than purifying phenol extracts, and our results support those presented in refs 29 and 30 in the manuscript, who find the same fluorescence component to represent the freshest portion of phenolic leachate extracts.

We have changed the word 'polyphenols' to 'phenols' throughout the manuscript to reflect the broader inclusion of both molecular classes as observed in the FTICR-MS data.

We have also added Supplementary Fig. S6 with details of the molecular characterization, and the following text to the Methods:

Lines 235-238: *“Component C5 (275/318 nm) was identified as a protein-like component that is associated with leaf litter phenol leachates^{28,29} (Supplementary Fig. S3), which we further confirmed with ultra-high resolution mass spectrometry data collected on a subset of samples in our PARAFAC model from an accompanying field-scale incubation (Supplementary Fig. S6).”*

We have also added Supplementary methods for “Molecular composition of “phenol” C5 component” (SI lines 21-41). This new analysis has resulted in the inclusion of two more co-authors and additional acknowledgements.

In addition, I found some more open tasks to be improved in this manuscript:

Introduction

1.94 mcrA gene copies

Why have the authors conducted analyses on DNA-level? DNA-level is just related to presence and may proliferation of methanogens but not for their activity. Much better would be to use mcrA transcripts or proteome analyses to estimate the activity.

Methane production is a decent and generally accepted measure of methanogen activity, and we have clarified this by adding text

on lines 105-111: *“Although relative abundance of the mcrA gene that we assayed does not entirely equate with specific activity of methanogen communities, there is strong evidence linking it with CH₄ production both here (i.e. Figs. 1-2) and in previous studies^{26,27}. This link arises because methanogenesis is not known to be a facultative process, but rather the only mechanism methanogens use to generate ATP (e.g. versus facultative denitrifiers in sediments and soils). A large methanogen population would typically only be sustained with concomitant rapid methane production rates.”*

Although the Reviewer suggests powerful and interesting tools that we could have used, we weren't really interested in transcriptional or post-translational control of the enzyme, per se. These processes wouldn't relate to overall activities any better than measuring the product/methane production (e.g. transcription would only primarily be active during binary fission; and allosteric inhibition could restrict activities of existing enzymes). Rather, given that methanogens are generally obligate methane producers, the simpler qPCR assay provides a decent proxy for their activity.

Material & Methods

First, please include the description of your control (CTR treatment) in your Experimental design.

Our description of the CTR treatment is given on lines 197-198. However we had neglected to refer to the CTR acronym, which made the link unclear. We have made this adjustment:

Lines 197-198: *"We also created replicated control jars (CTR) containing only base sediment, otherwise constructed and treated in the same manner."*

I.190 Please also show the CH₄ and CO₂ data from all incubation times! Is the rate of CH₄ emission also affected by the amended CON, DEC or TYP?

We have added Supplementary Fig. S5 showing the temporal pattern of CO₂ and CH₄ production and clarified our approach in the Methods:

Lines 206-213: *"We incubated the sediments and periodically collected headspace samples to measure CH₄ and CO₂ production after 150 days, representative of the length of a growing season in the Boreal Shield... We collected headspace gas by homogenizing 10 mL of N₂ into headspace prior to extracting a 10 mL gas sample by syringe, repeating this periodically over the 150-day incubation to ensure we reached a plateau of CO₂ and CH₄ production in all sediments (Supplementary Fig. S5)."*

The values presented are essentially rates, as they were all measured as total production over the same duration of 150 days. Thus, the rate of CH₄ production will mirror the absolute level of CH₄ produced.

I.209-213 I know that the analyses of polyphenols is not easy, but have the amended CON, DEC or TYP differed to each other in their composition and concentration of polyphenol? The composition and concentration of polyphenol should be different and the author should state that differences in the initial source of CON, DEC and TYP both before and after addition.

It is the highly bioavailable and soluble phenols that are most important, and while it would be nice to have a 'pre-addition' assessment of this material in the leaf litter, it is not clear how we would get this estimate. Our incubations started with leaf material in milliQ water. In lieu of a good way to estimate the leaching potential of a bioavailable fraction of phenols in the litter mix, we have measured the relative concentrations at the end of the incubation period. We have also added detailed information as to the molecular composition of this phenol component in Supplementary Fig. S6.

I.240/I.282/I.4 of supplement: 16S rRNA gene!

We thank the Reviewer for catching this and have made the change.

Figure 1: Please change spiked with addition of methanogen-rich sediment. Indicate the number of replicates.

The term 'methanogen-rich sediment' is used in the caption and we have now made this change in the legend. We have also added the number of replicates to the caption.

Lines 456-457: *"Different numbers (1-4) represent significant differences ($p < 0.05$) among amendments (ANOVA $F_{7, 24} = 39.47$), with $n=4$ replicates per amendment type."*

Figure 2: Why do the authors do not present absolute mcrA gene copy numbers per gram dw? Relative abundances are not meaningful here. Pooling of DNA is not helpful as now only technical replicates are shown. Line 438-439 is not clear for me. Where do the error bars come from?

In our case, bar plots of relative or absolute abundance would look identical just with a different scale. This is because the relative abundance values are related to absolute abundances per g dry weight, as they are divided by "abundance" of the control (i.e. if the control had 400 copies per gram and a treatment had 1200 copies per gram). So there is in effect, no difference in saying the relative abundance of the control is 1 and the treatment 3. The only downside is in the ability to directly compare to other qPCR studies, but we don't see this as affecting our interpretation of data, discussion, or conclusions in any meaningful way, so no changes were made.

As for why we still do not present copy numbers per gram dw, we were unable to obtain a high enough efficiency and R^2 with a methanogen pure culture standard curve for the mcrA gene fragment (the degenerate bases in our primer set appeared to cause the difficulty). Instead, we used a homogenized sample-PCR product mix that was diluted, better reflecting the variability and target sequences across our samples. This confirmed our reaction was valid but did not allow us to calculate true absolute abundance.

Finally, the error bars come from the three different %OM treatments (10, 20, and 40%), so we are comparing relative abundance across treatment types but not %OM. We pooled samples because we were low on template DNA following sequencing (and there were low initial yields from the extractions). However, treatment effects in CH_4 production were large between litter types, and little difference was seen between concentrations of any single litter type (see Figs 1-2). Therefore, to assay for relative methanogen abundance between litter types, we pooled DNA for each litter type regardless of concentration.

We have clarified the error bars in the figure caption on lines 466-467: *"Error bars for amendments represent standard error across %OM treatments (10, 20, 40%)."*

Supplementary methods

1.6 The authors should not pool sample to do statistics with biological replicates!

We pooled samples because we were low on template DNA and wanted to save some for mcrA qPCR and would not have been able to do both methanogen and SRB qPCR on fully replicated samples. We

agree that this is not ideal but we had multiple reasons to believe that chemical inhibition (i.e. by phenolic compounds) was the primary reason rates of methanogenesis were low. Nevertheless, we wanted to ensure that substrate competition by SRB (i.e. competitive inhibition of methanogens) in the DEC and CON treatments was not also a factor. Granted, we cannot comment on the variability of SRB numbers within treatments, but mean SRB numbers were very low (<1%) and consistent across all treatments. Therefore, it was reasonable to assume SRB were not playing an important biochemical role in the system, or at least that slow rates methanogenesis in the terrestrial litter treatments could not be explained by methanogens being out-competed for H₂ or CH₃COOH in favour of the thermodynamically more favourable sulfate reduction.

Supplementary Figure1: The authors should include the data of addition of methanogen-rich sediment. We have redone figure S1 to include the methanogen-rich (spiked) data, and also included the number of replicates in the caption as was suggested for Figure 1.

SI lines 64-68: *“Patterns of production in 10 and 40% OM methanogen-rich (spiked) and non-spiked sediments over a 150-day growing season are comparable to those seen in 20% OM (in Fig. 1). There are n=4 replicates per amendment type and results are shown on a log scale because of large differences between TYP and the other amendments.”*

REVIEWERS' COMMENTS:

Reviewer #1 (Remarks to the Author):

In this revision of the prior paper, I find the authors have clarified all of the issues that I brought up, some of which were errors on my part (for example, the latitudinal range of *Typha latifolia*). I think this paper is now ready to go, basically as is.

Reviewer #2 (Remarks to the Author):

The authors have satisfactorily responded to majority of my comments.

The study increases the understanding on CH₄ production in northern lakes. Nevertheless, there is a huge gap between the scales of the lab experiments and predictions across Boreal Shield resulting in multiplying of uncertainties. Estimates of CH₄ emissions that are based only on laboratory incubations of sediments slurries will result in large uncertainties in predicted increases in CH₄ production across the Boreal Shield. Overall CH₄ emissions from vegetation zones reflect also the gas dynamics of living plants, which can significantly contribute both to regional and global CH₄ emissions.

Reviewer #3 (Remarks to the Author):

Dear authors the revision was made nicely and clearly.

Only the figure legend of figure 2 and the supplementary methods should be slightly changed to my fully acceptance.

The authors has stated in their response "As for why we still do not present copy numbers per gram dw, we were unable to obtain a high enough efficiency and R² with a methanogen pure culture standard curve for the *mcrA* gene fragment (the degenerate bases in our primer set appeared to cause the difficulty). Instead, we used a homogenized sample-PCR product mix that was diluted, better reflecting the variability and target sequences across our samples. This confirmed our reaction was valid but did not allow us to calculate true absolute abundance."

Please add one sentence in the figure 2 for clarification of the readers.

In addition, the authors have stated "Granted, we cannot comment on the variability of SRB numbers within treatments, but mean SRB numbers were very low (<1%) and consistent across all treatments." in their response. this information should be also included in the supplementary information.

Emilson et al.

NCOMMS-17-22360A

We have indicated our responses in blue, and changes to manuscript text are shown in red. Changes are tracked in the revised manuscript. Line numbers in responses refer to line numbers in the revised document in Final format (Track Changes not shown).

REVIEWERS' COMMENTS:

Reviewer #1 (Remarks to the Author):

In this revision of the prior paper, I find the authors have clarified all of the issues that I brought up, some of which were errors on my part (for example, the latitudinal range of *Typha lattifolia*). I think this paper is now ready to go, basically as is.

No changes to the text have been made in response to this comment, as the Reviewer does not raise any new concerns. We thank the Reviewer again for their previous comments.

Reviewer #2 (Remarks to the Author):

The authors have satisfactorily responded to majority of my comments.

The study increases the understanding on CH₄ production in northern lakes. Nevertheless, there is a huge gap between the scales of the lab experiments and predictions across Boreal Shield resulting in multiplying of uncertainties. Estimates of CH₄ emissions that are based only on laboratory incubations of sediments slurries will result in large uncertainties in predicted increases in CH₄ production across the Boreal Shield. Overall CH₄ emissions from vegetation zones reflect also the gas dynamics of living plants, which can significantly contribute both to regional and global CH₄ emissions.

We agree with the Reviewer. We have reiterated the uncertainty and that the rates we present may be underestimates because of the processes they mentioned:

Lines 165-172: "Of course, these estimates are heavily caveated by several assumptions and uncertainties. For example, climate-driven changes in other factors, such as temperature, oxidation potential, and increased forest litterfall production, will certainly influence CH₄ production from lake sediments, and all production may not necessarily result in emissions¹. We have also not accounted for the gas dynamics of living plants, such as rhizosphere processes and aerenchymal transfer that may further enhance emissions where TYP is present¹¹. Similarly, we have not accounted for the differential mixing of forest-derived OM in sediments resulting from expected shifts in forest composition³⁴."

Reviewer #3 (Remarks to the Author):

Dear authors the revision was made nicely and clearly.

Only the figure legend of figure 2 and the supplementary methods should be slightly changed to my fully acceptance.

The authors has stated in their response "As for why we still do not present copy numbers per gram dw, we were unable to obtain a high enough efficiency and R2 with a methanogen pure culture standard curve for the mcrA gene fragment (the degenerate bases in our primer set appeared to cause the difficulty). Instead, we used a homogenized sample-PCR product mix that was diluted, better reflecting the variability and target sequences across our samples. This confirmed our reaction was valid but did not allow us to calculate true absolute abundance."

Please add one sentence in the figure 2 for clarification of the readers.

We have changed the figure caption as follows:

Lines: 505-512: "Fig. 2: Relative abundance of mcrA gene copies in amended sediments. Relative abundance is orders of magnitude higher in sediments amended with emergent macrophyte (*Typha latifolia*; TYP) litter than deciduous (DEC) or coniferous (CON) forest litter and mirrors CH₄ production in Fig. 1. DNA was pooled across replicates (n = 4 per %OM treatment) and expressed as relative abundance per gram dry-weight (gdw) of sediment normalized for extraction yield determined by qPCR. Samples were run in triplicate and compared to a standard curve generated from eDNA PCR product to capture the environmental variability in sequences. Error bars for amendments represent standard error across %OM treatments (10, 20, 40%)."

In addition, the authors have stated "Granted, we cannot comment on the variability of SRB numbers within treatments, but mean SRB numbers were very low (<1%) and consistent across all treatments." in their response. this information should be also included in the supplementary information.

We have added the following explanation to the Supplementary Table S2:

Lines 6-12 in Supplementary Information: "Supplementary Table 2: PCR results for sulphate reducing bacteria (SRB). PCR for the dsrA gene could not successfully amplify any detectable SRBs, and SRB related reads were in low abundance in the 16S rRNA gene sequencing libraries for each sediment type. Results from amended sediments are presented as means (SDs) across duplicates from pooled samples of three OM percentages (10, 20, 40%), with two control replicates (see methods). While pooling prevents comparison of variability within and among treatments, mean SRB numbers were consistently low (<1%) across treatments."